# Understanding catalytic synergy in dinuclear polymerization catalysts for sustainable polymers

Francesca Fiorentini[1], Wilfred T. Diment [1], Arron C. Deacy[1], Ryan W. F. Kerr [1], Stephen Faulkner[1] & Charlotte K. Williams [1] ✉

Understanding the chemistry underpinning intermetallic synergy and the discovery of generally applicable structure-performances relationships are major challenges in catalysis. Additionally, high-performance catalysts using earth-abundant, non-toxic and inexpensive elements must be prioritised. Here, a series of heterodinuclear catalysts of the form Co(III)M(I/II), where M(I/II) = Na(I), K(I), Ca(II), Sr(II), Ba(II) are evaluated for three different polymerizations, by assessment of rate constants, turn over frequencies, polymer selectivity and control. This allows for comparisons of performances both within and between catalysts containing Group I and II metals for $CO_2$/propene oxide ring-opening copolymerization (ROCOP), propene oxide/phthalic anhydride ROCOP and lactide ring-opening polymerization (ROP). The data reveal new structure-performance correlations that apply across all the different polymerizations: catalysts featuring s-block metals of lower Lewis acidity show higher rates and selectivity. The epoxide/heterocumulene ROCOPs both show exponential activity increases (vs. Lewis acidity, measured by the $pK_a$ of $[M(OH_2)_m]^{n+}$), whilst the lactide ROP activity and $CO_2$/epoxide selectivity show linear increases. Such clear structure-activity/selectivity correlations are very unusual, yet are fully rationalised by the polymerization mechanisms and the chemistry of the catalytic intermediates. The general applicability across three different polymerizations is significant for future exploitation of catalytic synergy and provides a framework to improve other catalysts.

Earth abundant element catalysis is essential to meet UN Sustainable Development Goals, and s-block metals are particularly attractive due to their abundance, low toxicity, diamagnetism, lack of redox chemistry and lack of color[1-6]. Catalytic synergy where two or more metals beneficially interact to enhance rates, selectivity and/or control is a useful strategy to improve their performances[6,7]. Synergic Group I and II metal catalysts have good precedent in homogeneous C-H activation, Grignard, hydro-elementation, Wittig and addition reactions, and as promoters in heterogeneous catalysis, e.g. nitrogen activation[1-5,8-10].

There are also a range of highly effective s-block metal and synergic polymerization catalysts[6,7,11-21]. Doi and Tonks, and co-workers, independently investigated olefin polymerization catalysts where alkalimetal cations coordinated adjacent to the transition metal active site enhanced performances[12,13]. Waymouth, Hedrick and co-workers discovered s-block thioureates that showed very high rates and excellent control in cyclic ester ring-opening polymerizations (ROP)[14,15]. Other Group I and/or II metal catalysts for lactide ROP have shown good rates and, in some cases, stereoselectivity[6,11,18-20]. Satoh and co-workers

[1]Department of Chemistry, University of Oxford, OX1 3TA Oxford, United Kingdom. ✉e-mail: charlotte.williams@chem.ox.ac.uk

showed that simple Group I carboxylates were effective catalysts for anhydride/epoxide ring-opening copolymerization (ROCOP) and switchable polymerizations[16,17].

Here, controlled polymerization catalysts are applied to make sustainable polyesters and polycarbonates. The resulting materials are useful plastics, elastomers, adhesives, fibers and surfactants[22–26]. Future sustainability is addressed by using monomers that are, or could be, derived from biomass, e.g. lactide (carbohydrate), phthalic anhydride (carbohydrate) and propene oxide (glycerol) and/or using waste $CO_2$[27,28]. All monomers are also existing commercial products, manufactured globally and already used in polymer manufacturing; hence the resulting polymers show real potential for large-scale applications. Polymerizations are atom-economical and products have end-life options including (mechanical) recycling, chemical recycling to monomer and, in some cases, biodegradation[22,23,29]. Three controlled and catalyzed polymerizations are widely applied to produce these polymers: *rac*-lactide (LA) ring-opening polymerization (ROP); phthalic anhydride (PA)/propene oxide (PO) ring-opening copolymerization (ROCOP); and $CO_2$/PO ring-opening copolymerization (ROCOP) (Fig. 1a, b)[22,25]. This work focusses on dinuclear catalysts which show strong precedent for high rates and selectivity in each of the processes[7,30–44]. Each polymerization operates by a mechanism involving metal alkoxide intermediates and although many (hundreds) of different catalysts are already reported for each separate polymerization, only a few are active for all three processes[25,34,45–48]. To accelerate future catalyst discovery, it would be very useful to identify general catalyst design rules transferable between different polymerizations, to understand structure-property relationships and to better exploit catalytic synergy.

Pioneering work from the groups of Coates, Nozaki and Lu, amongst others, demonstrated the significant potential of homodinuclear $CO_2$/epoxide ROCOP catalysts, mechanistic hypotheses proposed that two metals at optimum distances significantly enhance rates[34–36,41]. We, and others, reported heterodinuclear metal catalysts, some of which showed better performances than homodinuclear analogs[37,38,49–53]. However, only certain metal combinations were synergic[7,54]. Understanding structure-property relationships is very demanding and in many other cases there was no correlation between activity and any measurable catalyst structural parameter[38,52,53,55]. For example, the activities of a series of $CO_2$/epoxide ROCOP Mg(II)M(II) catalysts (M(II) = first row transition metals, Mg(II)) showed no correlation with trends in metallic ionic radii, bond dissociation energies, water exchange constants, electronegativity or oxophilicity[52]. Mashima and co-workers showed no correlation between the activities, for $CO_2$/epoxide ROCOP, of a series of tetranuclear Zn(II)Ln(III)$_3$ catalysts (Ln = lanthanide series) and trends in ionic radii, ligand exchange rates or bond dissociation energies[38]. In epoxide/anhydride ROCOP, synergic Al(III)M(I) catalysts were most active when M(I) = K(I), but the rationale for this metal partnership was not obvious[44]. In LA ROP, s-block metal catalysts often showed the highest rates when using Na(I) or K(I)[6,14–16,18,19,56–58]. Nonetheless, comparisons between Group I metal catalysts revealed contrasting trends: some authors proposed activity increased with ionic radius, whilst others hypothesized the exact opposite to rationalize experimental data[6,14–16,18,19,56–58].

## Results

Recently, we reported the first heterodinuclear Co(III)K(I) catalysts for PO/$CO_2$ ROCOP (Fig. 1c)[50]. Detailed kinetic and DFT analyses implicated a 'dinuclear metallate' mechanism featuring an anionic Co(III) and cationic K(I) catalyst and with a rate determining step involving epoxide ring opening by the carbonate nucleophile coordinated to the s-block metal (Supplementary Fig. 1)[51]. Here, two systematic series of Co(III)M(I/II) catalysts, where M(I/II) = Group I (Na(I), K(I), Rb(I), Cs(I)) or Group II (Ca(II), Sr(II), Ba(II)) are compared for PO/$CO_2$ ROCOP, PO/PA ROCOP, and *rac*-LA ROP, to investigate whether any general structure-performance relationships exist.

The series of catalysts were synthesized using a common route which involved reacting the dialdehyde precursor, Co(OAc)$_2$, and the Group I/II metal acetate, followed by the addition of ethylene diamine to form the Co(II)M(I/II) complexes, which were oxidized to the Co(III)M(I/II) catalysts with acetic acid, in air (Fig. 2)[50]. The catalysts were purified and isolated as brown powders in good yields (51–74%). The spectroscopic data ($^1H$, $^{13}C\{^1H\}$, COSY, HSQC, HMBC NMR and IR spectroscopy) are all consistent with the proposed structures, and purity of each complex was confirmed by elemental analysis (Supplementary Figs. 2–20)[8,50]. The complexes were also analyzed by single crystal X-ray diffraction which confirmed the heterodinuclear complexation and revealed 'cobaltate' structures, where the phenolate and acetate ligands are anionic donors (X-type) to the Co(III) center (Fig. 3, Supplementary Figs. S21 and S23)[51]. For the Co(III)M(II) series there are two bridging acetate ligands and one coordinated only to the M(II) center. All the complexes have intermetallic separations ~ 3–4 Å, a distance relevant to other dinuclear catalysts[21,36,49]. The solid-state structures highlight the different coordination chemistries between the s-block metals and the ligand. s-Block metals with smaller radii, like Ca(II) and Na(I), distort the ligand to minimize bond lengths, whilst larger cations, e.g. Rb(I), coordinate above the ligand-Co(III) plane and dimerize in the solid state. Metals in the medium size range, e.g. Sr(II), Ba(II), and K(I), coordinate within the ligand plane (Fig. 3, Supplementary Figs. S21–S27). The formation of cobaltate complexes is significant since

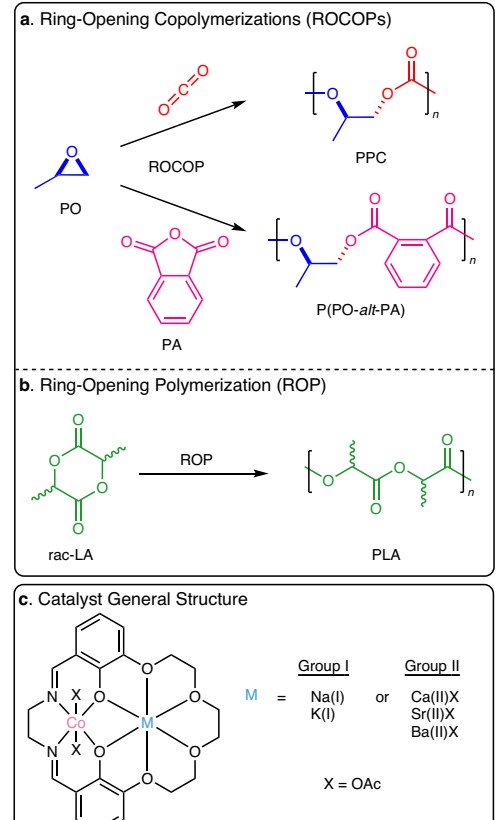

**Fig. 1 | The three polymerizations and the catalyst structures investigated. a** Scheme illustrating ring-opening copolymerization (ROCOP) of propene oxide (PO) and $CO_2$ to form poly(propene carbonate) (PPC); ROCOP of PO and phthalic anhydride (PA) to form or poly(PO-*alt*-PA). **b** Illustration of ring-opening polymerization (ROP) of *rac*-lactide (*rac*-LA) to form poly(lactide) (PLA); **c** Catalysts reported in this work (see Supplementary Information for details).

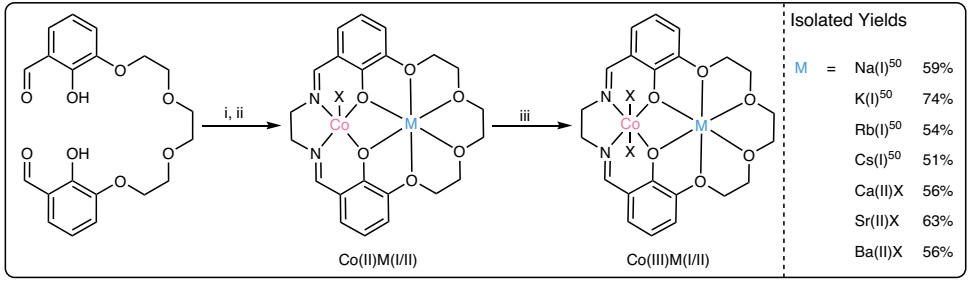

**Fig. 2 | Catalyst Synthesis.** Co(III)M(I/II) catalysts, M(I/III) = Na(I), K(I), Rb(I), Cs(I), Ca(II), Sr(II), Ba(II), with isolated yields (see supplementary information for experimental details). Reagents: i) Co(OAc)$_2$, M(I)(OAc) or M(II)(OAc)$_2$, MeCN, 1 h; ii) ethylene diamine, MeCN, 16 h; iii) Acetic acid, MeCN, air, 72 h[50].

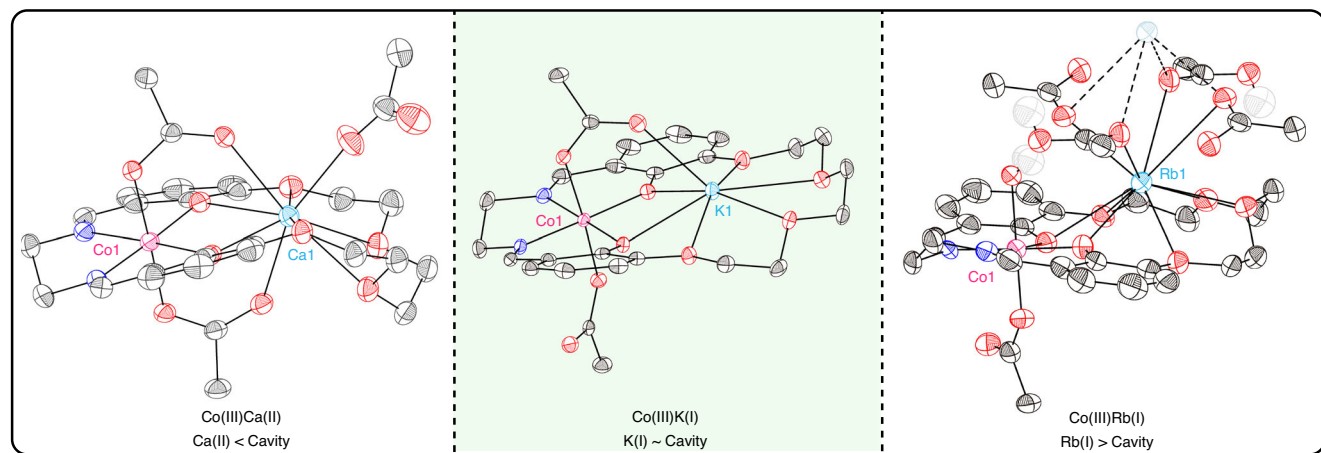

**Fig. 3 | Molecular structures of Co(III)Ca(II) (LHS), Co(III)K(I) (middle), and Co(III)Rb(I) (RHS) determined by X-ray diffraction (Supplementary Figs. 21–25).** The structures illustrate the structural differences between small, medium and large s-block metals and coordination to the ligand. Selected hydrogen atoms are omitted for clarity. Thermal ellipsoids are represented at 40% probability. Black spheres represent carbon atoms, red represent oxygen, dark blue represent nitrogen and pale gray represent hydrogen.

## Table 1 | Data for propene oxide (PO)/CO$_2$ ROCOP using the Co(III)M(I/II) catalysts, where PPC is poly(propene carbonate) and PC is propene carbonate

| Entry | Catalyst | Time/h | Productivity TON[a] | Activity TOF/h$^{-1}$[b] | Rate coefficient $k_{obs}$/ ×10$^{-3}$ s$^{-1}$[c] | Selectivity CO$_2$/%[d] | Selectivity PPC/%[e] | PPC $M_n$ [Đ]/ g mol$^{-1}$[f] |
|---|---|---|---|---|---|---|---|---|
| 1 | Co(III)Ca(II) | 22 | 194 ± 10 | 5 ± 1 | 0.36 ± 0.02 | 93 | 38 ± 4 | 700 [1.17] |
| 2 | Co(III)Sr(II) | 23 | 889 ± 44 | 23 ± 2 | 1.79 ± 0.14 | 97 | 50 ± 5 | 2100 [1.07] |
| 3 | Co(III)Ba(II) | 21 | 856 ± 43 | 30 ± 2 | 2.36 ± 0.17 | 97 | 61 ± 6 | 2500 [1.11] |
| 4[50] | Co(III)Na(I) | 5 | 600 ± 30 | 104 ± 8 | 7.52 ± 0.60 | >99 | 79 ± 2 | 2300 [1.08] |
| 5[50] | Co(III)K(I) | 4 | 1360 ± 68 | 528 ± 42 | 38.20 ± 3.06 | >99 | 98 ± 2 | 5900 [1.10] |
| 6[50] | Co(III)Rb(I) | 23 | 1240 ± 62 | 88 ± 7 | 6.37 ± 0.51 | >99 | 91 ± 3 | 6500 [1.07] |
| 7[50] | Co(III)Cs(I) | 23 | 1080 ± 54 | 88 ± 7 | 6.34 ± 0.51 | >99 | 84 ± 3 | 5600 [1.08] |

Reaction conditions: catalyst (0.025 mol%, 3.6 mM), PO (5 mL, 14.3 M), 1,2-cyclohexane diol (0.5 mol%, 71 mM), 20 bar CO$_2$, 50 °C.
[a]Turnover number (TON) = moles PO consumed/moles catalyst. PO conversion determined from $^1$H NMR spectra integrals for PPC (4.92 ppm), PC (4.77 ppm) and PPO (3.46-3.64 ppm), using mesitylene as an internal standard (0.05 mol%, 36 mM).
[b]Turnover frequency (TOF) are determined from 5 to 20% conversions, in all cases (Supplementary Tables 1 and 2, Supplementary Eq. (1)).
[c]Calculated from linear ln([PO]$_t$/[PO]$_0$) vs. time plots (Supplementary Figs. 28 and 30).
[d]Selectivity for CO$_2$ conversion, determined from PO conversion to PPC + PC/overall PO conversion (using NMR integrals).
[e]Selectivity for PPC formation, determined as PO conversion to PPC/overall PO conversion (using $^1$H NMR spectroscopy integrals).
[f]Determined by GPC, in THF, at 30 °C, using narrow MW polystyrene calibrants.

DFT investigations implicated this structure is present in the active site during catalysis[50,51].

All the new catalysts were tested for PO/CO$_2$ ROCOP under common conditions: in neat epoxide using 1:4000, [catalyst]:[PO]$_0$ (0.025 mol% catalyst), at 20 bar CO$_2$, 50 °C and with 20 equivalents vs. catalyst of 1,2-cyclohexane diol (CHD, 0.5 mol%) (Table 1). These conditions are selected to allow comparisons against some of the best performing catalysts in the field and represent 'demanding' loadings[50,51]. The diol, CHD, controls the polycarbonate molar mass and dispersity; it is used here to target polyols with $M_n$ < 6000 g/mol which are relevant to polyurethane, surfactant and resin applications[24]. Catalysts often show reduced rates when using excess diol, therefore identification of high activity synergic catalysts is important[50,59,60].

All catalysts were active but there are clear differences with turnover frequencies (TOFs) ranging from 5 to 588 h$^{-1}$ depending on the metal combination. For all complexes, the catalytic rates are determined per mole of complex since DFT investigations indicated that

**Table 2 | Data for propene oxide (PO)/phthalic anhydride (PA) ROCOP with Co(III)M(I/II) catalysts**

| Entry | Catalyst | Time/h | Productivity TON[a] | Activity TOF/h[-1b] | Rate coefficient $k_{obs}$/×$10^{-6}$ M$^{-1}$ s$^{-1c}$ | Selectivity polyester/%[d] | Polyester $M_n$ [Đ]/g mol$^{-1e}$ |
|---|---|---|---|---|---|---|---|
| 1 | Co(III)Ca(II) | 21 | 88 ± 2 | 4 ± 2 | 17.4 ± 7.4 | >99 | 4600 [1.13]<br>11600 [1.02] |
| 2 | Co(III)Sr(II) | 30 | 87 ± 1 | 3 ± 1 | 10.5 ± 3.2 | >99 | 5100 [1.08] |
| 3 | Co(III)Ba(II) | 30 | 84 ± 1 | 3 ± 1 | 11.2 ± 3.6 | >99 | 6000 [1.11] |
| 4 | Co(III)Na(I) | 7 | 88 ± 2 | 12 ± 1 | 45.8 ± 1.6 | >99 | 4200 [1.12] |
| 5 | Co(III)K(I) | 3 | 89 ± 1 | 31 ± 1 | 122.7 ± 4.2 | >99 | 4500 [1.11] |

Reaction conditions: Catalyst (0.1 mol%, 0.0143 mM), PA (10 mol%, 1.43 mM), PO (1 mL, 14.3 M), 50 °C (Catalyst:PA:PO, 1:100:1000).
[a]TON determined by $^1$H NMR spectroscopy using integrals for PA (8.10–7.85 ppm) and P(PO-alt-PA) (7.75–7.40 ppm).
[b]Turnover frequency (TOF) is reported from gradients of conversion vs. time plots, from 5 to 20% conversions, in all cases (Supplementary Tables 3 and 4, Supplementary Eq. (2)).
[c]Rate coefficient is the gradient of linear fits to plots of [PA]t vs. time (Supplementary Fig. 36).
[d]Selectivity for polyester determined by $^1$H NMR spectroscopy using integrals for P(PO-alt-PA) and any PPO (3.46–3.64 ppm).
[e]Determined by GPC, in THF, at 30 °C, using narrow dispersity polystyrene calibrants.

only one acetate group initiates polymerization[51]. The catalysts also showed variable selectivity for polycarbonate vs. cyclic carbonate formation, with values between 38 and 98% (Table 1). In all cases, the uptake of $CO_2$ was very high, from 93 to 99%, resulting in the formation of highly alternating polycarbonates. All the polycarbonates showed narrow, monomodal molar mass distributions (Đ ≤ 1.17); these features signal controlled polymerization catalysis (Supplementary Figs. 31 and 33)[22,29]. Generally, the catalysts featuring Group I metals, M(I), were more active and selective than those with Group II metals, M(II). It is immediately clear that catalyst performance rationale by 's-block metal ion size' are inappropriate since Co(III)K(I) shows the best activity and selectivity, yet Co(III)Ba(II) or Co(III)Sr(II) feature cations of similar sizes and stronger ligand binding affinities yet are significantly worse catalysts (Supplementary Figs. 34 and 35)[61].

Epoxide/anhydride ROCOP is an attractive route to polyesters with variable structures and properties, due to the wide range of polymerizable monomers, many of which are commercial products and/or bio-based[29,62]. It yields semi-aromatic, rigid, aliphatic and/or functionalised polyesters and, in contrast to cyclic ester ROP, shows favorable thermodynamics, allowing high yields of polyesters[29,62]. Given its utility, better understanding of catalyst design parameters are urgently needed. The Co(III)M(I/II) catalysts were evaluated using commercial, and widely used, monomers; PO and PA. Polymerizations were conducted in neat PO, using loadings of 1:100:1000 [catalyst]:[PA]$_0$:[PO]$_0$ (i.e. 0.1 mol% catalyst, 10 mol% PA) and at 50 °C (Table 2). Once again, there were clear differences in catalytic activity, with Co(III)K(I) being more active than the Na(I) analog, and both being faster than M(II) catalysts. Plots of P(PO-alt-PA) molar mass against anhydride conversion were linear, and the dispersity values remained narrow throughout the reactions (Supplementary Figs. 37 and 46). These features are all consistent with well-controlled polymerizations and single site catalysts; important factors in any structure-activity relationship investigation. The Co(III)K(I) catalyst was also highly active and selective using a range of other epoxides and anhydrides to produce different polyesters (Supplementary Table 5, Supplementary Figs. 47 and 55).

Polylactide (PLA) is the leading bio-derived polymer used in packaging, fiber and plastics applications[22,26,63]. It has lower greenhouse gas emissions than incumbent fossil-plastics since it is plant-derived, and, after use, PLA is recyclable and/or compostable[22,26,63]. LA ROP is used to produce PLA, and its efficiency depends upon catalyst selection[11]. The catalysts were tested using 1:100:1000 [catalyst]$_0$:[LA]$_0$:[PO]$_0$ (i.e. 0.1 mol% catalyst), at 50 °C. The PO serves as both reaction solvent and initiator, since it reacts with the catalyst to form the true metal-alkoxide 'initiators' (acetate ligands cannot initiate)[25]. When monitoring the reaction only LA ROP occurred; no ether linkages from PO ROP were observed in the PLA.

The most active catalyst was Co(III)K(I), achieving a TOF of 912 h$^{-1}$, and in all cases the Co(III)M(I) catalysts outperform the Co(III)M(II)

species (Table 3). The catalysts all show good polymerization control, as evidenced by monomodal molar mass distributions and linear plots of molar mass vs. conversion (Supplementary Figs. 61 and 70). The PLA was atactic except for the most active Co(III)K(I) catalyst which showed moderate iso-selectivity ($P_i$ = 0.71, Supplementary Table 8). Comparing the M(I) and M(II) catalysts shows that activity trends do not correlate with the s-block cationic radius. For example, Co(III)K(I) and Co(III)Ba(II) have very similar radii (K(I) = 1.46 Å, Ba(II) = 1.52 Å), yet the K(I) catalyst is 6 times more active than the Ba(II) analog (Table 3; Supplementary Fig. 71)[64].

## Discussion

To understand the experimental trends in activity and selectivity data, the polymerization mechanisms should be discussed. As mentioned, comparing M(I) vs. M(II) catalysts reveals that rationale based only on ionic radii are incomplete or inaccurate. There was also no correlation between rates or selectivity and published s-block cation binding affinity to 18-crown-6, or ionic radius data (Supplementary Figs. 34, 35, 69). Thus, our attention turned to evaluating the s-block metal Lewis acidity, since the rate determining step in all three polymerizations involves a metal-oxygenated nucleophile, proposed as coordinated to the s-block metal[19,49–51,54]. Prior work has revealed that metal Lewis acidity can be accurately quantified using the p$K_a$ of the metal aqua complex, in water[65]. Recently, Blakemore and Kumar reported an in-depth investigation of s-block cation Lewis acidity, analyzing metal triflates dissolved in organic solvents by Guttmann Beckett $^{31}$P NMR titration methods[65]. In all solvents tested, plots of s-block cation p$K_a$ vs. the change in $^{31}$P NMR chemical shift of triphenyl phosphine oxide, used as a titrant, were linear. This confirms that p$K_a$ can be used to quantify Lewis acidity in organic media[65].

In the current series of catalysts, the plot of PO/$CO_2$ ROCOP activity against s-block metal p$K_a$ value showed exponential increase, i.e. the fastest catalysts featured the least Lewis acidic metals, Na(I) and K(I) (Fig. 4a). The exponential fits were confirmed by linear fits to ln(TOF) vs. ln(p$K_a$) and ln($k_{obs}$) vs. ln(p$K_a$) plots (Supplementary Figs. 73 and 74). Analysis of the selectivity for polycarbonate (PPC) vs. s-block metal Lewis acidity (p$K_a$) showed a linear correlation with the most selective catalysts featuring the least Lewis acidic s-block metals, Na(I) and K(I) (Fig. 4c). Such metal Lewis acidity-performance correlations in $CO_2$/epoxide ROCOP are so far without precedent yet should be very useful for future catalyst design.

Polymerization mechanisms that were previously established using kinetics, DFT and reactivity investigations are used to rationalize the trends[50,51]. The Co(III)M(I) catalysts showed second-order rate laws, dependent upon both catalyst and epoxide concentrations[50,51]. Rates were zeroth order in $CO_2$ pressure from 10 to 40 bar[50,51]. The 'dinuclear metallate' mechanism involves a rate-determining step where PO is coordinated at Co(III) and is ring-opened by a K(I)-carbonate (Fig. 4e, Supplementary Fig. S1)[50,51]. The selectivity-determining step controls

**Table 3 | Data for *rac*-lactide (LA) ring opening polymerization using the Co(III)M(I/II) catalysts**

| Entry | Catalyst | Time/h | TON[a] | TOF/h$^{-1b}$ | $k_{obs}$/×10$^{-3}$ s$^{-1c}$ | $M_n$ [Đ] / g mol$^{-1d}$ |
|---|---|---|---|---|---|---|
| 1 | Co(III)Ca(II) | 4.5 | 90 | 42 ± 4 | 0.21 ± 0.02 | 3300 [1.22] |
| 2 | Co(III)Sr(II) | 1.7 | 97 ± 1 | 212 ± 12 | 1.04 ± 0.06 | 4700 [1.23] |
| 3 | Co(III)Ba(II) | 1.7 | 97 ± 1 | 147 ± 10 | 0.72 ± 0.05 | 5600 [1.61] |
| 4 | Co(III)Na(I) | 0.2 | 88 ± 1 | 670 ± 22 | 3.29 ± 0.21 | 4100 [1.30] |
| 5 | Co(III)K(I) | 0.2 | 90 ± 4 | 912 ± 83 | 4.48 ± 0.41 | 4800 [1.23] |

Reaction conditions: catalyst (0.1 mol%, 14.3 mM), *rac*-LA (10 mol%, 1.43 M), PO (1 mL, 14.3 M), 50 °C ([Catalyst]$_0$:[LA]$_0$:[PO]$_0$, 1:100:1000).
[a]TON determined by $^1$H NMR spectroscopy from normalized integrals for *rac*-LA (4.99–5.09 ppm, 2 H) and PLA (5.10–5.26 ppm, 2H).
[b]Turnover frequency (TOF) determined from 5 to 70% conversion (Supplementary Tables 6 and 7, Supplementary Eq. (S3)).
[c]Calculated from linear fits to plots of ln([*rac*-LA]$_t$/[*rac*-LA]$_0$) vs. time for Co(III)Na(I) and Co(III)K(I); for Co(III)Ca(II), Co(III)Sr(II) and Co(III)Ba(II) slow initiation was handled using [PLA] = A(1 − exp(−$k_{obs}$t)) (Supplementary Figs. 56 and 60)[71].
[d]Determined by GPC in THF, at 30 °C, using narrow dispersity polystyrene standards and multiplied by 0.58 as the correction factor[72].

the relative quantities of polycarbonate vs. cyclic carbonate formed. This step is pre-rate determining and equilibrates a Co(III)K(I)-alkoxide (II) with $CO_2$ to form the Co(III)K(I)-carbonate (III); the equilibrium is controlled by the $CO_2$ insertion chemistry[50,51]. The by-product, cyclic carbonate, forms only from the alkoxide intermediate and so reducing its relative concentration increases selectivity for polycarbonate.

For the Co(III)M(I/II) catalysts, rates were greater with weakly Lewis acidic s-block metals. It is proposed that reducing the metal Lewis acidity slightly destabilises the metal-carbonate intermediates (i.e. higher ground state energy) and may also reduce transition state energies during epoxide ring-opening. The selectivity for polycarbonate also increases as s-block metal Lewis acidity decreases. By the same rationale, the least acidic s-block metals feature slightly destabilized alkoxide intermediates (i.e. higher ground state energy). DFT calculations suggested that the alkoxide intermediate is stabilized by interaction with an adjacent carbonate group on the growing polymer chain (II, Fig. 4d, e). As the s-block metals' Lewis acidity decreases, this interaction weakens, which further destabilizes the alkoxide intermediate and drives $CO_2$ insertion to increase the formation of the carbonate intermediate. The Co(III)Rb(I) and Co(III)Cs(I) catalysts do not fit the activity or selectivity trends and were not tested in other polymerizations. These complexes feature cations of sufficiently large ionic radii that it limits effective coordination within the macrocycle, as observed in the solid-state structures, and results in aggregation. These aggregates complicate structure-activity correlations (Rb(I) = 1.56 Å, Cs(I) = 1.71 Å, vs. K = 1.46 Å) (Supplementary Figs. 75 and 76)[50,64].

Analysis of the data and trends for the series of catalysts in PO/PA ROCOP reveals a related activity trend to that observed for PO/$CO_2$ ROCOP. The activity, as assessed by either rate constant or turnover-frequency, increases exponentially as the Lewis acidity decreases (Fig. 5a). It is very interesting to note that once again the rates increase exponentially vs. metal cation p$K_a$ value (linear plots of ln(TOF) vs. ln(p$K_a$) and ln($k_{obs}$) vs. ln(p$K_a$), Supplementary Figs. 77 and 78). For these catalysts, the rate law for epoxide/anhydride ROCOP is first order in epoxide and zero-order in anhydride concentration, respectively (Supplementary Figs. 79 and 80)[44,66]. A related Al(III)K(I) catalyst showed a similar rate law and was proposed to operate by a related 'metallate' mechanism by DFT calculations[44]. In this series of catalysts, the proposed rate determining step involves epoxide coordination at Co(III) and ring-opening by M(I)- or M(II)-carboxylate intermediates.

The experimental data show faster rates with less Lewis acidic s-block metals. This data is rationalized by the least Lewis acidic s-block metals having slightly destabilized, and hence more reactive, carboxylate intermediates, compared to those with more Lewis acidic metals. The same exponential increases to activity vs. p$K_a$ data for the two different epoxide/heteroallene ROCOP is fully consistent with the polymerizations having related rate-determining steps. Finally, both polymerizations ROCOPs involve rate determining steps

where nucleophilic attack occurs from bidentate nucleophiles, either carbonate ($CO_2$/epoxide) or carboxylate (anhydride/epoxide) intermediates. It is tentatively proposed that these complexes may show exponential rate vs. acidity relationships since these key s-block metal intermediates can accommodate both monodentate and bidentate coordination modes, as observed in the solid-state structures. Comparing the two polymerizations, the activities for PO/PA ROCOP are considerably lower than for the equivalent polymerizations using $CO_2$ likely due to the lower nucleophilicity of the M(I/II) carboxylate vs. carbonate in the rate determining step. Once again 'model' p$K_a$ values help to rationalize the rate differences since formic acid (p$K_a$ = 3.75) is considerably more acidic than carbonic acid (p$K_a$ = 6.37) and weaker acids are shown, in this work, to be more reactive[67].

In the series of catalysts, the rates of LA ROP, as assessed either from rate constants ($k_{obs}$) or TOFs, increase linearly with decreasing s-block metal Lewis acidity (Fig. 6a). The coordination-insertion mechanism for LA ROP is proposed in common with many other metal-based lactide polymerization catalysts[6,11]. The polymerization kinetics are all first order in lactide and the rate-limiting step is proposed to involve metal-alkoxide intermediate attack at (metal-)coordinated lactide (Fig. 6c, Supplementary Figs. S56 and S57). It is proposed that the lactide coordinates at the Co(III) site and the M(I/II)-alkoxide attacks it – with the metals exhibiting similar roles to those in epoxide/heteroallene ROCOP. The experimental data trends are rationalized since decreasing the s-block Lewis acidity results in a destabilized metal-alkoxide intermediate which shows higher lability in the lactide insertion reaction (and faster rates). The different activity vs. s-block metal Lewis acidity data trends reveal new information to better understand catalyst design parameters: intermediates involving bidentate species, such as carboxylates or carbonates, show exponential increases in activity, whereas those which involve monodentate ligands, such as alkoxides, show linear trends. Accordingly, the PO/$CO_2$ selectivity and lactide rates show linear correlations and depend upon the reactivity of metal-alkoxide intermediates (Fig. 5d). There was also a linear correlation between the degree of isotacticity, $P_i$, and s-block metal Lewis acidity consistent with the alkoxide intermediate controlling stereoselectivity (Supplementary Fig. 81).

Overall, the series of Co(III)M(I/II) catalysts show unprecedented but reproducible correlations between both catalytic activity and selectivity (for polymer or stereoselectivity) and the s-block metal Lewis acidity. These correlations apply equivalently between three different polymerizations, which are all relevant to the future increased production of sustainable polyesters and polycarbonates. The findings are significant since they signal clearly the methods to optimize performances and provide improved understanding of the physical-chemical factors responsible for high rates/selectivity. Reducing the Lewis acidity of the s-block metal benefits both the rates and selectivity of these polymerizations, with the Co(III)K(I) catalyst consistently performing best. In the future, research should target ancillary ligand modifications to fine-tune K(I) Lewis acidity, for example by

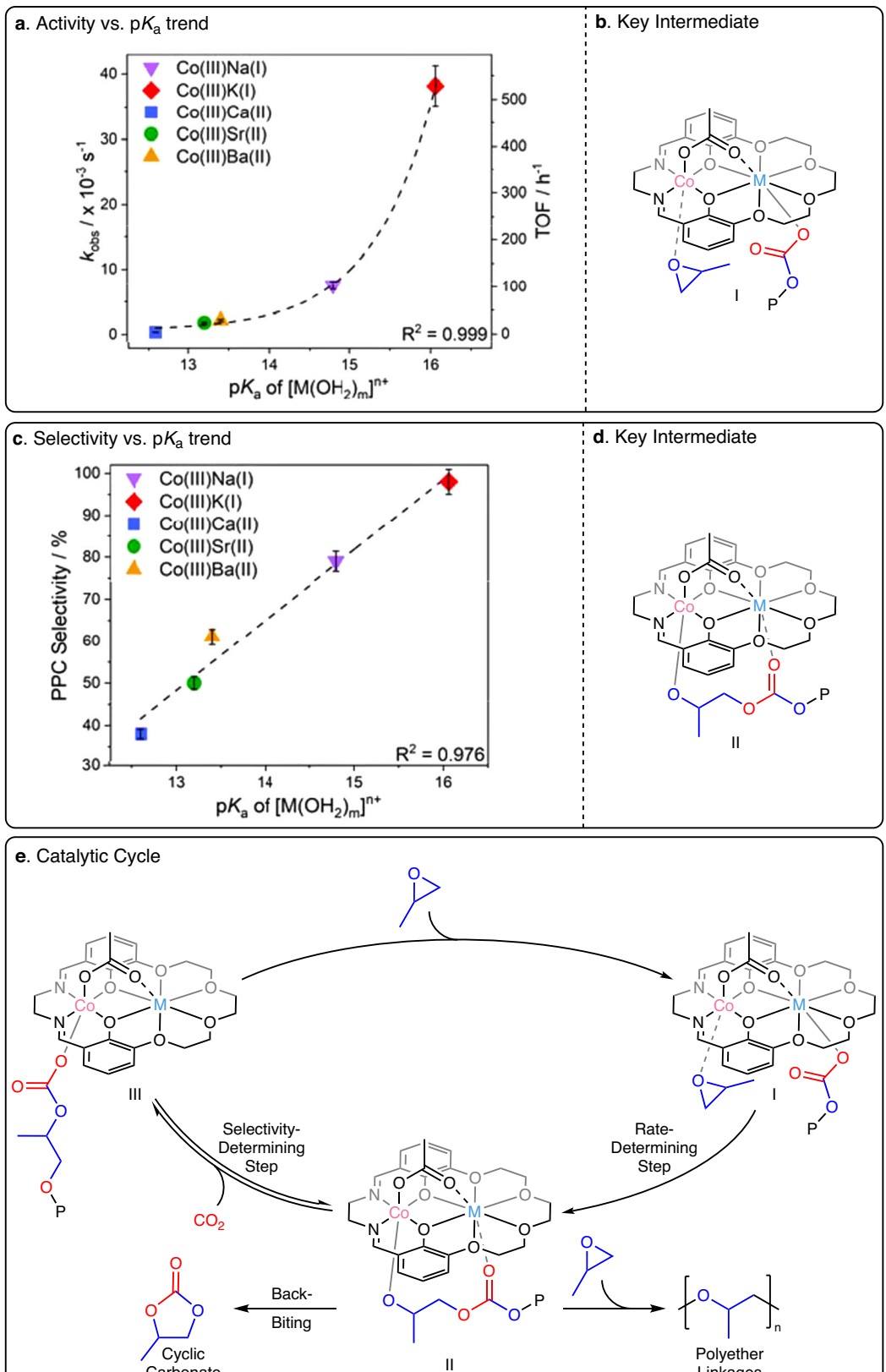

**Fig. 4 | Structure-activity and structure-selectivity trends for propene oxide and carbon dioxide ROCOP, with the catalytic cycle. a** Plots of $k_{obs}$ and TOF against the $pK_a$ of $[M(OH_2)_m]^{n+}$ for PO/$CO_2$ ROCOP catalyzed by Co(III)M(I/II); error bars represent the standard error of the mean; **b** Structure of the proposed rate determining intermediate; **c** Plot of selectivity against s-block metal $pK_a$ value; error bars represent the standard error of the mean; **d** Structure of the selectivity determining intermediate; **e** Catalytic cycle for PO/$CO_2$ ROCOP catalyzed by Co(III) M(I/II) with rate and selectivity determining steps identified. P denotes the polymer chain.

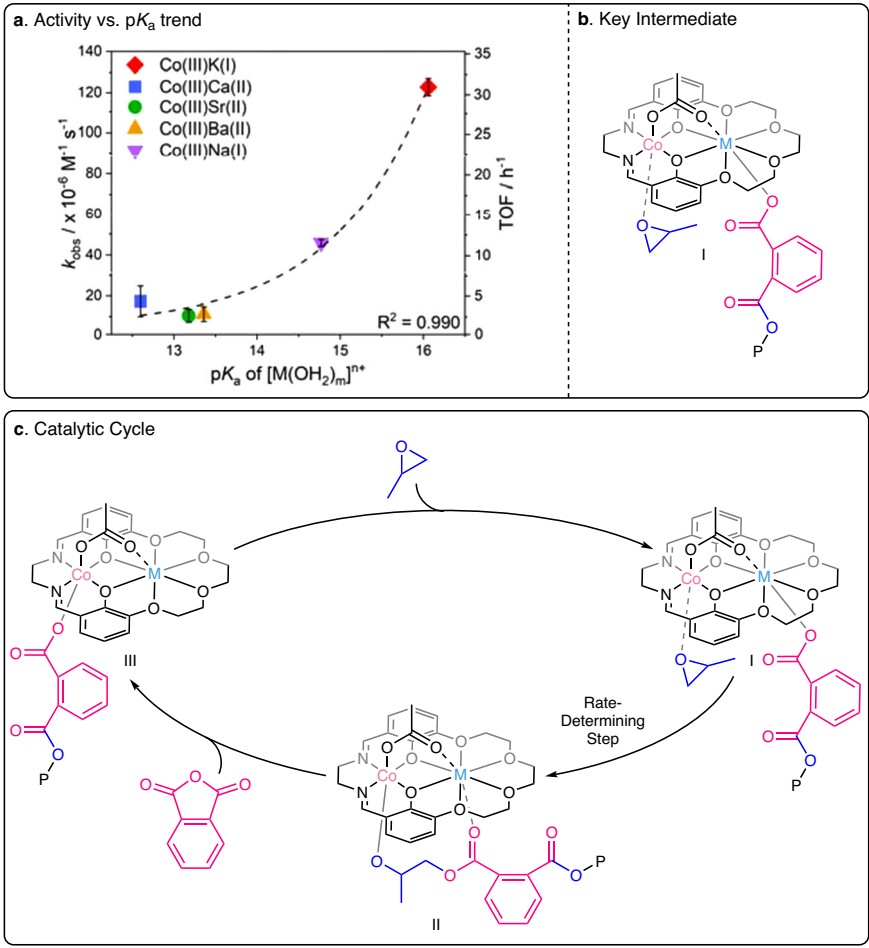

**Fig. 5 | Structure-activity trends for propene oxide and phthalic anhydride ROCOP, with the catalytic cycle. a** Plot of $k_{obs}$ and TOF against p$K_a$ of the s-block metal for PO/PA ROCOP catalyzed by Co(III)M(I/II); error bars represent the standard error of the mean; **b** Key intermediate that determines activity; **c** Catalytic cycle for PO/PA ROCOP catalyzed by Co(III)M(I/II). P denotes the polymer chain.

altering the substituents adjacent to the 'ether' donors and by investigating different numbers of ether donors. The trends uncovered for these heterodinuclear catalysts should be applicable to other polymerization catalysts including those based on simple s-block metal cations and synergic catalysts with other metals. It is recommended that in future all high-performance s-block metal catalysts are assessed using p$K_a$ as a measure of Lewis acidity and that Group II metals catalysts are routinely compared with Group I analogs[6,18]. In synergic dinuclear catalysis, replacing an expensive second transition metal or even obviating phosphonium co-catalyst by ligand design to incorporate s-block metals should be prioritized as a means to rationally re-design metal Schiff-base and salen catalysts for heteroallene/epoxide ROCOP. The Co(III)K(I) catalysts show excellent rates and selectivity in three different polymerizations and so could be highly effective using monomer mixtures, following rules of switchable catalysis, to produce multi-block polymers and phase-separated materials[25]. The structure-performance relationships uncovered in this work are expected to apply to other synergic s-block metal catalysts for organic transformations, polymerizations and may even shed light on roles played by s-block cation 'promoters' which are very common in heterogeneous catalysis[9,68,69]. There are even di-zinc histone deacetylase enzymes activated by K(I) cations and these Lewis acidity trends may also aid understanding of synergic bio-catalysts[70]. Finally, the activation and utilization of $CO_2$ is of central importance and the current structure-performance trends should help to reduce both the cost and weight of metals applied and guide mechanistic understanding of $CO_2$ 'insertions' and transformations.

A series of heterodinuclear Co(III)M complexes, where M = Group I or II cations, were compared as polymerization catalysts for three processes relevant to future sustainable polymer production: carbon dioxide/epoxide ring-opening copolymerization, anhydride/epoxide ring-opening copolymerization and *rac*-lactide ring-opening polymerization. Each polymerization data set showed clear structure-activity and structure-selectivity trends where the best catalysts feature the least Lewis acidic s-block metals, i.e. Co(III) and Na(I) or K(I). Importantly, the same activity and selectivity trends vs. cation Lewis acidity applied to all three different polymerizations. Such clear structure-performance correlations are very rare in any polymerization catalysis and are without precedent across these different reactions. The activity and selectivity data show clearly that the s-block metal-oxygenate nucleophiles become more reactive when using less Lewis acidic metals. For s-block metal carbonate or carboxylate intermediates rate data increased exponentially, whilst for s-block metal alkoxide intermediates rate or selectivity data increased linearly. The data and correlations help rationalize the beneficial roles played by s-block metals and the chemistry underpinning catalytic metal synergy; they should accelerate future discovery of higher activity, selectivity and controlled catalysts. These are needed to unlock production of sustainable, bio-based and recyclable polymers. In future, the best catalysts should also be investigated for polymerizations of monomer mixtures (e.g. LA, $CO_2$, PA and PO), exploiting mechanism switches, to efficiently and selectively produce block and copolymers[25]. Such materials diversify the properties and applications for materials made from biomass and $CO_2$[25].

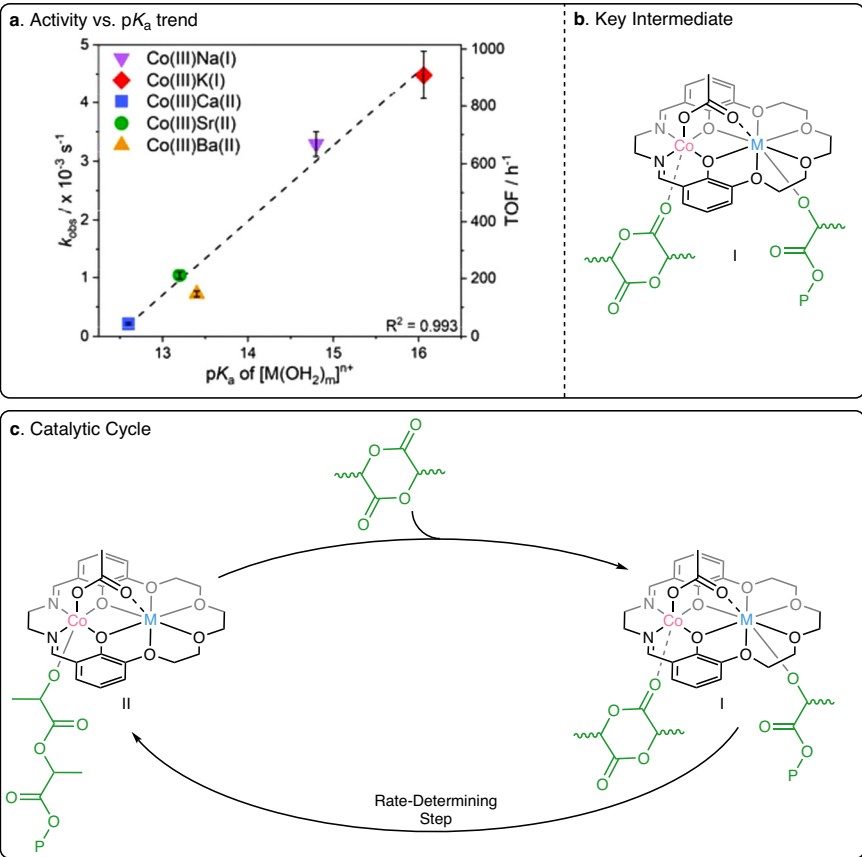

**Fig. 6 | Structure-activity trend for racemic lactide ROP, with the catalytic cycle. a** Plot of $k_{obs}$ and TOF against p$K_a$ of the s-block metal for *rac*-LA ROP catalyzed by Co(III)M(I/II); error bars represent the standard error of the mean; **b** Proposed structure of the intermediate that determines activity; **c** Catalytic cycle for *rac*-LA ROP catalyzed by Co(III)M(I/II). P denotes the polymer chain.

## Data availability

The electronic supplementary information provides data to support the manuscript, all other data are available from the corresponding author. The X-ray crystallographic coordinates for structures reported in this study have been deposited at the Cambridge Crystallographic Data Center (CCDC), under deposition numbers 2250536, 2250537 and 2250538. These data can be obtained free of charge from The Cambridge Crystallographic Data Center via www.ccdc.cam.ac.uk/data_request/cif.

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

## Acknowledgements

The EPSRC (EP/S018603/1; EP/R027129/1), Research England (RED, RE-P-2020-04) and the OxICFM (Inorganic Chemistry for Future Manufacturing) CDT (EP/S023828/1) (F.F.) are acknowledged for funding.

## Author contributions

F.F., W.D. and A.D. conceived and conducted all experiments; F.F. and R.K. performed all crystallographic characterization. C.W. and S.F. conceived the project, led the research team, and secured the research funding.

## Competing interests

C.W. is a director of Econic Technologies. The other authors declare no competing interests.
