## [Peer Review File · Nature Communications]

Understanding Catalytic Synergy in Dinuclear Polymerization Catalysts for Sustainable PolymersReviewers' Comments:

Reviewer #1:

Remarks to the Author:

This manuscript containing comparative data for three polymerization processes critical to sustainable technology is a true masterpiece of catalytic studies. It clearly demonstrates the same structure-activity and structure-selectivity for the three quite different polymerization processes. The presentation is well-written and referenced, providing an excellent understanding of the synergy exhibited by these dinuclear catalysts. I found the manuscript to be easy to follow with very good figures, therefore I can honestly offer no suggestions for improving its quality. Publish in its present form.

Reviewer #2:

Remarks to the Author:

A question I have had as an expert in inorganic chemistry and polymer synthesis is counter cation effect especially in the reactions performed using group 1-2 metal based alkoxides or carboxylate. Normally, chemists consider it simply as an idle counter ion for actively working alkoxide or carboxylate anions but some significant effect has been often observed by variation of the 1-2 metal cations, which has not been well understood. In this work, a series of bimetallic complexes composed of Co and group 1-2 metals were prepared and effect of the variation of group 1-2 metals was clearly revealed with polymerization data gained very nicely in a series of CO₂/propene oxide ring-opening copolymerization, propene oxide/phthalic anhydride, and lactide ring-opening polymerization; Lewis acidity of the main group metals (PKa values) is quantitatively correlated with the activity and selectivity! I read this manuscript with enthusiasm, gaining new information and insight in designing catalyst and especially in selection of counter cation of group 1-2 metals and I think many other scientists and researchers also gain the same benefits as I by reading this paper, strongly recommending for publication. Paper was also well written; few flaws and errors are found in reading.

- 1) I recommend to move the activity and rate constant columns just right side of TON column in Tables 1-2. Reporting TON values in Tables 1-3 needs writing the reaction times in Table footnote or head.
- 2) Footnote "b" is missing in Table 2 which should be corrected.
- 3) To make clear "0.58 as the correction factor" in footnote in Table 3 is recommended to be changed to "multiplied by 0.58 as the correction factor".
- 4) The term "PKa values of s-block cations" is ambiguous and incorrect (e.g., PKa value of Na⁺?). Many chemists cannot understand this term. I can understand the term through reading the reference cited: "Lewis acidity strength estimated by PKa values of [M(OH₂)_m]ⁿ⁺". Please rephrase the term correctly in the main text as well as in Figures, making readers understand without reading the paper cited. Statement about "s-block cation pKa vs. ³¹P NMR chemical shifts" is out of the blue either and cannot be also understood without reading the paper cited. Simple changing "³¹P NMR chemical shift" to "³¹P NMR chemical shift of Ph₃P=O added as a titrant" may help readers understand.

Understanding Catalytic Synergy in Dinuclear Polymerization Catalysts for Sustainable Polymers

Francesca Fiorentini, Wilfred T. Diment, Arron C. Deacy, Ryan W. F. Kerr, Stephen Faulkner, Charlotte K. Williams*

Reviewer 1:

This manuscript containing comparative data for three polymerization processes critical to sustainable technology is a true masterpiece of catalytic studies. It clearly demonstrates the same structure-activity and structure-selectivity for the three quite different polymerization processes. The presentation is well-written and referenced, providing an excellent understanding of the synergy exhibited by these dinuclear catalysts. I found the manuscript to be easy to follow with very good figures, therefore I can honestly offer no suggestions for improving its quality. Publish in its present form.

We thank reviewer 1 for the review and interest in the work.

Reviewer 2:

A question I have had as an expert in inorganic chemistry and polymer synthesis is counter cation effect especially in the reactions performed using group 1-2 metal based alkoxides or carboxylate. Normally, chemists consider it simply as an idle counter ion for actively working alkoxide or carboxylate anions but some significant effect has been often observed by variation of the 1-2 metal cations, which has not been well understood. In this work, a series of bimetallic complexes composed of Co and group 1-2 metals were prepared and effect of the variation of group 1-2 metals was clearly revealed with polymerization data gained very nicely in a series of CO₂/propene oxide ring-opening copolymerization, propene oxide/phthalic anhydride, and lactide ring-opening polymerization; Lewis acidity of the main group metals (pK_a values) is quantitatively correlated with the activity and selectivity! I read this manuscript with enthusiasm, gaining new information and insight in designing catalyst and especially in selection of counter cation of group 1-2 metals and I think many other scientists and researchers also gain the same benefits as I by reading this paper, strongly recommending for publication. Paper was also well written; few flaws and errors are found in reading.

Thank you – the review is helpful and we have been able to address all comments.

1) I recommend to move the activity and rate constant columns just right side of TON column in Tables 1-2. Reporting TON values in Tables 1-3 needs writing the reaction times in Table footnote or head.

We agree and have now adjusted Tables 1 – 3 to include a column for the reaction times and also re-ordered the Table columns as per the suggestion.

2) Footnot "b" is missing in Table 2 which should be corrected.

Thank you – this has been corrected and Table 2 now contains 'b' (for activity).

3) To make clear "0.58 as the correction factor" in footnote in Table 3 is recommended to be changed to "multiplied by 0.58 as the correction factor".

Thank you, the footnote for Table 3 now reads as suggested (with reference to the primary literature).

4) The term "PKa values of s-block cations" is ambiguous and incorrect (e.g., PKa value of Na+?). Many chemists cannot understand this term. I can understand the term through reading the reference cited: "Lewis acidity strength estimated by PKa values of $[M(OH_2)_m]^{n+}$ ". Please rephrase the term correctly in the main text as well as in Figures, making readers understand without reading the paper cited. Statement about "s-block cation pKa vs. ^{31}P NMR chemical shifts" is out of the blue either and cannot be also understood without reading the paper cited. Simple changing " ^{31}P NMR chemical shift" to " ^{31}P NMR chemical shift of $Ph_3P=O$ added as a titrant" may help readers understand.

We agree that the terminology and how it is used in this work should be clarified. The x-axes of relevant plots have been changed from "pKa of s-block metal" to "pKa of $[M(OH_2)_m]^{n+}$ " (Figures 4A, 4C, 5A, 6A). The main text has also been clarified (highlighted in yellow in the text):

Page 1, lines 12-14: "The epoxide/heterocumulene ROCOPs both show exponential activity increases (vs. Lewis acidity, measured by the pKa of $[M(OH_2)_m]^{n+}$), whilst the lactide ROP activity and CO_2 /epoxide selectivity show linear increases."

Page 5, lines 6-7: "Prior work has revealed that metal Lewis acidity can be accurately quantified using the pKa of the metal aqua complex, in water."

Page 5, lines 8-10: "In all solvents tested, plots the pKa of the s-block metal aqua complex vs. the change in ^{31}P NMR chemical shift of triphenyl phosphine oxide, used as a titrant, were linear."